# SUBSPACE REGULARIZERS FOR FEW-SHOT CLASS INCREMENTAL LEARNING

**Afra Feyza Akyürek**
Boston University
akyurek@bu.edu

**Ekin Akyürek**
MIT CSAIL
akyurek@mit.edu

**Derry Tanti Wijaya**
Boston University
wijaya@bu.edu

**Jacob Andreas**
MIT CSAIL
jda@mit.edu

## ABSTRACT

Few-shot class incremental learning—the problem of updating a trained classifier to discriminate among an expanded set of classes with limited labeled data—is a key challenge for machine learning systems deployed in non-stationary environments. Existing approaches to the problem rely on complex model architectures and training procedures that are difficult to tune and re-use. In this paper, we present an extremely simple approach that enables the use of ordinary logistic regression classifiers for few-shot incremental learning. The key to this approach is a new family of *subspace regularization* schemes that encourage weight vectors for new classes to lie close to the subspace spanned by the weights of existing classes. When combined with pretrained convolutional feature extractors, logistic regression models trained with subspace regularization outperform specialized, state-of-the-art approaches to few-shot incremental image classification by up to 23% on the *mini*ImageNet dataset. Because of its simplicity, subspace regularization can be straightforwardly configured to incorporate additional background information about the new classes (including class names and descriptions specified in natural language); this offers additional control over the trade-off between existing and new classes. Our results show that simple geometric regularization of class representations offers an effective tool for continual learning.[1]

## 1    INTRODUCTION

Standard approaches to classification in machine learning assume a fixed training dataset and a fixed set of class labels. But for many real-world classification problems, these assumptions are unrealistic. Classifiers must sometimes be updated on-the-fly to recognize new concepts (e.g. new skills in personal assistants or new road signs in self-driving vehicles), while training data is sometimes unavailable for reuse (e.g. due to privacy regulations, Lesort et al. 2019; McClure et al. 2018; or storage and retraining costs, Bender et al. 2021). Development of models that support **few-shot class-incremental learning** (FSCIL), in which classifiers' label sets can be easily extended with small numbers of new examples and no retraining, is a key challenge for machine learning systems deployed in the real world (Masana et al., 2020).

As a concrete example, consider the classification problem depicted in Fig. 1. A model, initially trained on a large set of examples from several **base classes** (*snorkel*, *arctic fox*, *meerkat*; Fig. 1a), must subsequently be updated to additionally recognize two **novel classes** (*white wolf* and *poncho*; Fig. 1b), and ultimately distinguish among all five classes (Fig. 1c). Training a model to recognize the base classes is straightforward: for example, we can jointly optimize the parameters of a feature extractor (perhaps a convolutional network parameterized by $\theta$) and a linear classification layer ($\eta$) to maximize the regularized likelihood of (image, label) pairs from the dataset in Fig. 1a:

$$\mathcal{L}(\theta, \eta) = \frac{1}{n} \sum_{(x,y)} \log \frac{\exp(\eta_y^\top f_\theta(x))}{\sum_{y'} \exp(\eta_{y'}^\top f_\theta(x))} + \alpha \left( \|\eta\|^2 + \|\theta\|^2 \right) \qquad (1)$$

But how can this model be *updated* to additionally recognize the classes in Fig. 1b, with only a few examples of each new class and no access to the original training data?

---

[1]Code for the experiments is released under https://github.com/feyzaakyurek/subspace-reg.

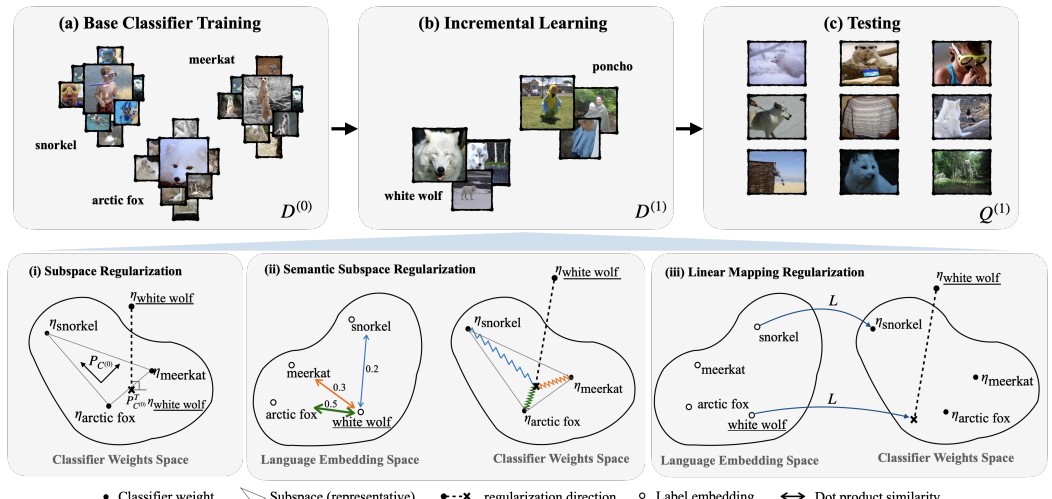

Figure 1: Few-shot class incremental learning: (a) A *base classifier* is trained on a large dataset ($D^{(0)}$). (b) This classifier is extended to also discriminate among a set of new classes with a small number of labeled examples ($D^{(1)}$). (c) Models are evaluated on a test set that includes all seen classes ($Q^{(1)}$). This paper focuses on extremely simple, regularization-based approaches to FSCIL, with and without side information from natural language: (i) We regularize novel classifier weights toward the shortest direction to the subspace spanned by base classifier weights. (ii) We regularize novel classifiers pulling them toward the weighted average of base classifiers where weights are calculated using label/description similarity between novel and base class names or one-sentence descriptions. (iii) We learn a linear mapping $L$ between word labels and classifier weights of the base classes. Later, we project the novel label *white wolf* and *regularize* the novel classifier weight $\eta_{\text{white wolf}}$ towards the projection.

Naïvely continuing to optimize Eq. 1 on (x, y) pairs drawn from the new dataset will cause several problems. In the absence of any positive examples of those classes, performance on *base classes* will suffer due to **catastrophic forgetting** (Goodfellow et al., 2013), while performance on *novel classes* will likely be poor as a result of **overfitting** (Anderson & Burnham, 2004).

As a consequence, most past work on FSCIL has focused on alternative approaches that use non-standard prediction architectures (e.g., Tao et al., 2020b) or optimize non-likelihood objectives (e.g., Yoon et al., 2020; Ren et al., 2019). This divergence between approaches to standard and incremental classification has its own costs—state-of-the-art approaches to FSCIL are complicated, requiring nested optimizers, complex data structures, and numerous hyperparameters. When improved representation learning and optimization techniques are developed for standard classification problems, it is often unclear to how to apply these to the incremental setting.

In this paper, we turn the standard approach to classification into a surprisingly effective tool for FSCIL. Specifically, we show that both catastrophic forgetting and overfitting can be reduced by introducing an additional **subspace regularizer** (related to one studied by Agarwal et al. 2010 and Kirkpatrick et al. 2017) that encourages novel $\eta$ to lie close to the subspace spanned by the base classes. On its own, the proposed subspace regularizer produces ordinary linear classifiers that achieve state-of-the-art results on FSCIL, improving over existing work in multiple tasks and datasets.

Because of its simplicity, this regularization approach can be easily configured to incorporate additional information about relationships between base and novel classes. Using language data as a source of background knowledge about classes, we describe a variation of our approach, which we term **semantic subspace regularization**, that pulls weight vectors toward particular convex combinations of base classes that capture their semantic similarity to existing classes. Semantic subspace regularization results in a comparable performance to subspace regularization on average; it better preserves performance on existing classes at the expense of higher accuracy on newer classes. These results suggest that FSCIL and related problems may not require specialized machinery to solve, and that simple regularization approaches can solve the problems that result from limited access to training data for both base and novel classes.

## 2 BACKGROUND

A long line of research has focused on the development of automated decision-making systems that support online expansion of the set of concepts they can recognize and generate. An early example (closely related to our learning-from-definitions experiment in Section 5) appears in the classic SHRDLU language grounding environment (Winograd, 1972): given the definition *a steeple is a small triangle on top of a tall rectangle*, SHRDLU acquires the ability to answer questions containing the novel concept *steeple*. Recent work in machine learning describes several versions of this problem in featuring more complex perception or control:

**Few-shot and incremental learning** *Few-shot* classification problems test learners' ability to distinguish among a fixed set of classes using only a handful of labeled examples per class (Scheirer et al., 2012). Most effective approaches to few-shot learning rely on additional data for pre-training (Tian et al., 2020) or meta-learning (Vinyals et al., 2016; Finn et al., 2017; Snell et al., 2017; Yoon et al., 2019). One peculiarity of this evaluation paradigm is that, even when pre-trained, models are evaluated only on new (few-shot) classes, and free to update their parameters in ways that cause them to perform poorly on pre-training tasks. As noted by past work (Kuzborskij et al., 2013), a more realistic evaluation of models' ability to rapidly acquire new concepts should consider their ability to discriminate among both new concepts and old ones, a problem usually referred to as *few-shot class-incremental learning* (FSCIL)[2] (Tao et al., 2020b).

FSCIL requires learners to incrementally acquire novel classes with few labeled examples while retaining high accuracy on previously learned classes. It combines the most challenging aspects of *class-incremental learning* (Rebuffi et al., 2017) *task-incremental learning* (Delange et al., 2021), and *rehearsal-based learning* (Rolnick et al., 2019; Chaudhry et al., 2019), three related problems with much stronger assumptions about the kind of information available to learners. Existing approaches to this problem either prioritize novel class adaptation (Ren et al., 2019; Yoon et al., 2020; Chen & Lee, 2021; Cheraghian et al., 2021) or reducing forgetting in old classes (Tao et al., 2020b).

**Learning class representations** Even prior to the widespread use of deep representation learning approaches, the view of classification as problem of learning *class* representations motivated a number of approaches to multi-class and multi-task learning (Argyriou et al., 2007a; Agarwal et al., 2010). In few-shot and incremental learning settings, many recent approaches have also focused on the space of class representations (Tao et al., 2020a). Qi et al. (2018) initialize novel class representations using the average features from few-shot samples. Others (Gidaris & Komodakis, 2018; Yoon et al., 2020; Zhang et al., 2021) train a class representation predictor via meta-learning, and Tao et al. (2020b) impose topological constraints on the manifold of class representations as new representations are added. Alternatively, Chen & Lee (2021) models the visual feature space as a Gaussian mixture and use the cluster centers in a similarity-based classification scheme. Lastly, two concurrent works condition both old and new class representations at each session according to an auxiliary scheme; graph attention network in Zhang et al. (2021) and relation projection in Zhu et al. (2021).

Our approach is related to Ren et al. (2019), who uses a nested optimization framework to *learn* auxiliary parameters for every base and novel class to influence the novel weights via regularization; we show that these regularization targets can be derived geometrically without the need for an inner optimization step. Also related is the work of Barzilai & Crammer (2015), which synthesizes the novel weights as linear combinations of base weights; we adopt a regularization approach that allows learning of class representations that are not strict linear combinations of base classes. Moreover, Kuzborskij et al. (2013) study a class incremental learning setup where they increment the number of classes by one. Similar to ours, the parameters for the novel class is regularized towards a weighted combination of old class parameters while using as many examples from old classes as there are from novel classes. In comparison, our approach does not require any examples from old classes.

**Learning with side information from language** The use of background information from other modalities (especially language) to bootstrap learning of new classes is widely studied (Frome et al.,

---

[2]Variants of this problem have gone by numerous names in past work, including *generalized few-shot learning* (Schönfeld et al., 2019), *dynamic few-shot learning* (Gidaris & Komodakis, 2018) or simply *incremental few-shot learning* (Ren et al., 2019; Chen & Lee, 2021).

2013; Radford et al., 2021; Reed et al., 2016; Yan et al., 2021)—particularly in the **zero-shot learning** and **generalized zero-shot learning** where side information is the *only* source of information about the novel class (Chang et al., 2008; Larochelle et al., 2008; Akata et al., 2013; Pourpanah et al., 2020). Specialized approaches exist for integrating side information in few-shot learning settings (Schwartz et al., 2019; Cheraghian et al., 2021).

## 3 PROBLEM FORMULATION

We follow the notation in Tao et al. (2020b) for FSCIL: assume a stream of $T$ **learning sessions**, each associated with a labeled dataset $D^{(0)}, D^{(1)}, \ldots, D^{(T)}$. Every $D^{(t)}$ consists of a **support set** $S^{(t)}$ (used for training) and a **query set** $Q^{(t)}$ (used for evaluation). We will refer to the classes represented in $D^{(0)}$ as **base classes**; as in Fig. 1a, we will assume that it contains a large number of examples for every class. $D^{(1)}$ (and subsequent datasets) introduce **novel classes** (Fig. 1b). Let $C(S) = \{y : (x, y) \in S\}$ denote the set of classes expressed in a set of examples $S$; we will write $C^{(t)} = C(S^{(t)})$ and $C^{(\leq t)} := \bigcup_{j \leq t} C(S^{(j)})$ for convenience. The learning problem we study is *incremental* in the sense that each support set contains only new classes ($C^{(t)} \cap C^{(<t)} = \emptyset$)[3], while each query set evaluates models on both novel classes and previously seen ones ($C(Q^{(t)}) = C^{(\leq t)}$). It is *few-shot* in the sense that for $t > 0$, $|S^{(t)}|$ is small (containing 1–5 examples for all datasets studied in this paper). Given an incremental learning session $t > 0$ the goal is to *fine-tune* existing classifier with the limited training data from novel classes such that the classifier performs well in classifying all classes learned thus far.

**FSCIL with a single session** Prior to Tao et al. (2020b), a simpler version of the multi-session FSCIL was proposed by Qi et al. (2018) where there is only single incremental learning session after the pre-training stage i.e. $T = 1$. This version, which we call *single-session* FSCIL, has been extensively studied by previous work (Qi et al., 2018; Gidaris & Komodakis, 2018; Ren et al., 2019; Yoon et al., 2020). This problem formulation is the same as above with $T = 1$: a feature extractor is trained on the samples from $D^{(0)}$, then $D^{(1)}$, then evaluated on samples with classes in $C^{(0)} \cup C^{(1)}$.

## 4 APPROACH

Our approach to FSCIL consists of two steps. In the base session, we jointly train a feature extractor and classification layer on base classes (Section 4.1). In subsequent (incremental learning) sessions, we freeze the feature extractor and update only the classification layer using regularizers that (1) stabilize representations of base classes, and (2) bring the representations of new classes close to existing ones (Sections 4.2-4.4).

### 4.1 FEATURE EXTRACTOR TRAINING

As in Eq. 1, we begin by training an ordinary classifier comprising a non-linear feature extractor $f_\theta$ and a linear decision layer with parameters $\eta$. We choose $\eta$ and $\theta$ to maximize:

$$\mathcal{L}(\eta, \theta) = \frac{1}{|S^{(0)}|} \sum_{(x,y) \in S^{(0)}} \log \frac{\exp(\eta_y^\top f_\theta(x))}{\sum_{c \in C^{(0)}} \exp(\eta_c^\top f_\theta(x))} - \alpha \left( \|\eta\|^2 + \|\theta\|^2 \right) \qquad (2)$$

As discussed in Section 5, all experiments in this paper implement $f_\theta$ as a convolutional neural network. In subsequent loss formulations we refer to $\|\eta\|^2 + \|\theta\|^2$ as $R_{\text{prior}}(\eta, \theta)$.

### 4.2 FINE-TUNING

Along with the estimated $\hat{\theta}$, feature extractor training yields parameters only for base classes $\eta_{y \in C^{(0)}}$. Given an incremental learning dataset $D^{(t)}$, we introduce new weight vectors $\eta_{c \in C^{(t)}}$

---

[3]This is the original setup established by Tao et al. (2020b). We will also present experiments in which we retain one example per class for memory replay following Chen & Lee (2021).

and optimize

$$\mathcal{L}(\eta) = \frac{1}{|S^{(t)}|} \sum_{(x,y) \in S^{(t)}} \log \frac{\exp(\eta_y^\top f_{\hat{\theta}}(x))}{\sum\limits_{c \in C^{(\leq t)}} \exp(\eta_c^\top f_{\hat{\theta}}(x))} - \alpha R_{\text{prior}}(\eta, \mathbf{0}) - \beta R_{\text{old}}^{(t)}(\eta) - \gamma R_{\text{new}}^{(t)}(\eta) . \quad (3)$$

with respect to $\eta$ alone. Eq. 3 features two new regularization terms, $R_{\text{old}}^{(t)}$ and $R_{\text{new}}^{(t)}$. $R_{old}^t$ which also appears in past work (Kuzborskij et al., 2013), limits the extent to which fine-tuning can change parameters for classes that have already been learned:

$$R_{\text{old}}^{(t)}(\eta) = \sum_{t' < t} \sum_{c \in C^{(t')}} \|\eta_c^{t'} - \eta_c\|^2 \quad (4)$$

where $\eta_c^{t'}$ denotes the value of the corresponding variable at the end of session $t'$. (For example, $\eta_c^0$ refers to the weights for the base class $c$ prior to fine tuning, i.e. after session $t' = 0$.) As shown in Section 5.2, using $R_{\text{old}}$ alone, and setting $R_{\text{new}} = 0$, is a surprisingly effective baseline; however, performance can be improved by appropriately regularizing new parameters as described below.

**Variant: Memory**   Following past work (Chen & Lee, 2021) which performs incremental learning while retaining a small "memory" of previous samples $M$, we explore an alternative baseline approach in which we append $S^{(t)}$ in Eq. 3 with $M^{(t)}$. We define the memory at session $t$ as $M^{(t)} = \bigcup_{(t' < t)} M^{(t')}$ where $M^{(t')} \subseteq S^{(t')}$ and $|M^{(t')}| = |C^{(t')}|$. We sample only 1 example per previous class and we reuse the same example in subsequent sessions.

### 4.3   METHOD 1: SUBSPACE REGULARIZATION

Past work on other multitask learning problems has demonstrated the effectiveness of constraining parameters for related tasks to be similar (Jacob et al., 2008), lie on the same manifold (Agarwal et al., 2010) or even on the same linear subspace (Argyriou et al., 2007a). Moreover, Schönfeld et al. (2019) showed that a shared latent feature space for all classes is useful for class-incremental classification. Features independently learned for novel classes from small numbers of examples are likely to capture spurious correlations (unrelated to the true causal structure of the prediction problem) as a result of dataset biases (Arjovsky et al., 2019). In contrast, we expect most informative *semantic* features to be shared across multiple classes: indeed, cognitive research suggests that in humans' early visual cortex, representations of different objects occupy a common feature space (Kriegeskorte et al., 2008). Therefore, regularizing toward the space spanned by base class weight vectors encourages new class representations to depend on semantic rather than spurious features and features for all tasks to lie in the same universal subspace.

We apply this intuition to FSCIL via a simple subspace regularization approach. Given a parameter for an incremental class $\eta_c$ and base class parameters $\{\eta_{j \in C^{(0)}}\}$, we first compute the **subspace target** $m_c$ for each class. We then compute the distance between $\eta_c$ from $m_c$ and define:

$$R_{\text{new}}^{(t)}(\eta) = \sum_{c \in C^{(t)}} \|\eta_c - m_c\|^2 \quad (5)$$

where $m_c$ is the projection of $\eta_c$ onto the space spanned by $\{\eta_{j \in C^{(0)}}\}$:

$$m_c = P_{C^{(0)}}^\top \eta_c \quad (6)$$

and $P_{C^{(0)}}$ contains the orthogonal basis vectors of the subspace spanned by the initial set of base weights $\eta_{j \in C^{(0)}}$. ($P_{C^{(0)}}$ can be found using a QR decomposition of the matrix of base class vectors, as described in the appendix.)

Previous work that leverages subspace regularization for multitask learning assume that data from all tasks are available from the beginning (Argyriou et al., 2007b; Agarwal et al., 2010; Argyriou et al., 2007a). Our approach to subspace regularization removes these assumptions, enabling tasks (in this case, novel classes) to arrive incrementally and predictions to be made cumulatively over all classes seen thus far without any further information on which task that a query belongs to. Agarwal et al. (2010) is similar to ours in encouraging all task parameters to lie on the same manifold; it is different in that they learn the manifold and the task parameters alternately. Also related Simon et al. (2020) and Devos & Grossglauser (2019) model class representations over a set of subspaces (disjoint in the latter) for non-incremental few-shot learning.

## 4.4 METHOD 2: SEMANTIC SUBSPACE REGULARIZATION

The constraint in Eq. 5 makes explicit use of geometric information about base classes, pulling novel weights toward the base subspace. However, it provides no information about *where* within that subspace the weights for a new class should lie—potentially causing interference with base classes. In most classification problems, classes have names consisting of natural language words or phrases; these names often contain a significant amount of information relevant to the classification problem of interest. (Even without having ever seen a *white wolf*, a typical English speaker can guess that a *white wolf* is more likely to resemble an *arctic fox* than a *snorkel*.) These kinds of relations are often captured by *embeddings* of class labels (or more detailed class descriptions) (Pennington et al., 2014).

When available, this kind of information about class semantics can be used to construct an improved subspace regularizer by encouraging new class representations to lie close to a convex combination of base classes weighted by their semantic similarity. We replace the subspace projection $P_{C^{(0)}}^{\top} \eta_c$ in Eq. 5 with a **semantic target** $l_c$ for each class. Letting $e_c$ denote a semantic embedding of the class $c$, we compute:

$$R_{\text{new}}^{(t)}(\eta) = \sum_{c \in C^{(t)}} \| \eta_c - l_c \|^2 \tag{7}$$

where

$$l_c = \sum_{j \in C^{(0)}} \frac{\exp\left(e_j \cdot e_c / \tau\right)}{\sum_{j \in C^{(0)}} \exp\left(e_j \cdot e_c / \tau\right)} \eta_j \tag{8}$$

and $\tau$ is a hyper-parameter. Embeddings $e_c$ can be derived from multiple sources: in addition to the class names discussed above, a popular source of side information for zero-shot and few-shot learning problems is *detailed textual descriptions* of classes; we evaluate both label and description embeddings in Section 5.

Schönfeld et al. (2019) also leverage label information on a shared subspace for few-shot incremental learning where they project both visual and semantic features onto a shared latent space for prediction in the single-session setting. In comparison, we re-use the base visual space for joint projection for multiple incremental sessions.

**Baseline: Linear Mapping** While the approach described in Eq. 7 combines semantic information and label subspace information, a number of previous studies in vision and language have also investigated the effectiveness of directly learning a mapping from the space of semantic embeddings to the space of class weights (Das & Lee, 2019; Socher et al., 2013; Pourpanah et al., 2020; Romera-Paredes & Torr, 2015). Despite pervasiveness of the idea in other domains, this is the first time we are aware of it being explored for FSCIL. We extend our approach to incorporate this past work by learning a linear map $L$ between the embedding space $e_j \in E$ and the weight space containing $\eta_{C^{(0)}}$:

$$L^* = \min_{L} \sum_{j \in C^{(0)}} \| \eta_j - L(e_j) \|^2 \tag{9}$$

then set

$$R_{\text{new}}^{(t)} = \sum_{c \in C^{(t)}} \| \eta_c - L^*(e_c) \|^2 . \tag{10}$$

Concurrent work by (Cheraghian et al., 2021) also leverages side information for FSCIL where they learn a mapping *from* image space *onto* the label space to directly produce predictions in the label space. We provide comparisons in Section 5.

## 5 EXPERIMENTS

Given a classifier trained on an initial set of base classes, our experiments aim to evaluate the effect of subspace regularization (1) on the learning of new classes, and (2) on the retention of base classes. To evaluate the generality of our method, we evaluate using two different experimental paradigms that have been used in past work: a *multi-session* experiment in which new classes are continuously

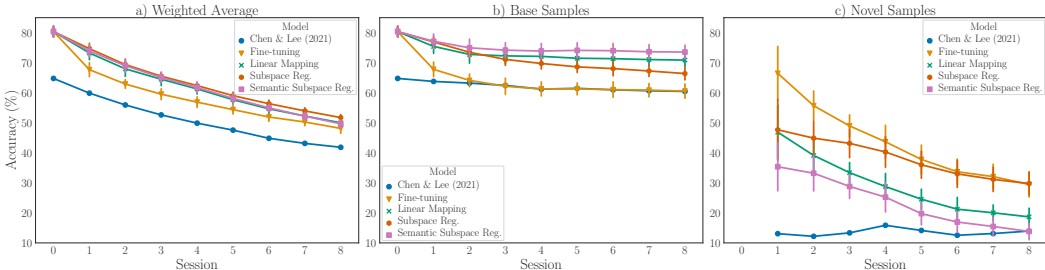

Figure 2: Multi-Session FSCIL accuracy (%) results on *mini*ImageNet. In the first session *0*, there are a total of 60 classes (*base*). Every session following the first one introduces 5 *novel* classes with 5 labeled samples from each. Each session provides accuracy over all classes that were seen thus far. Weighted average is the weighted combination of novel and base accuracies with respect to the number of classes in each category. Error bars are standard deviation (95% CI). In accordance with Chen & Lee (2021) we preserve only one sample per class from previous classes and append them to the support set during fine-tuning (+M variant). Regularization based approaches i.e. *subspace regularization*, *semantic subspace regularization* and *linear mapping* consistently outperform previous benchmark and *fine-tuning* in weighted average (a).

added and the classifier must be repeatedly updated, and a *single-session* setup ($T = 1$) in which new classes arrive only once. As our optimizer, we use stochastic gradient descent for feature extractor training and gradient descent in incremental sessions to train all models. Details about experiment setups and results are discussed below. Additional details may be found in the appendix.

## 5.1 MULTI-SESSION

We follow the same setup established in Tao et al. (2020b) as well as Section 3: we first train a ResNet (He et al., 2016) network from scratch as the feature extractor on a large number of examples from base classes $C^{(0)}$ to obtain an initial classifier $\eta_{j \in C^{(0)}}$. We then observe a new batch of examples $S^{(t)}$ and produce a new classifier defined by $\eta_{c \in C^{(\leq t)}}$. Finally, we evaluate the classifier according to *top-1* accuracy in base and novel samples as well as their weighted average (Tao et al., 2020b; Chen & Lee, 2021). We use the *mini*ImageNet dataset (Vinyals et al., 2016; Russakovsky et al., 2015) for our multi-session evaluation. *mini*ImageNet contains 100 classes with 600 samples per class.

Table 1: Multi-Session FSCIL weighted average of accuracy (%) results on *mini*ImageNet using an identical setup to Fig. 2 with memory distinction. We report the average results over 10 random splits of the data for incremental sessions 1, 2 and 8. $\pm M$ indicates 1 sample per class is kept (or not) in the memory to further regularize forgetting. Our Fine-tuning baseline is already superior to previous results. In both memory settings, our regularizers substantially outperform respective benchmarks for all 1-8 sessions. *Results are only estimates from the plot in the respective work. Bold indicates the highest.

| Session | 1 | | 2 | | 8 | |
| Model | $-M$ | $+M$ | $-M$ | $+M$ | $-M$ | $+M$ |
|---|---|---|---|---|---|---|
| Tao et al. (2020b) | 50.1 | | 45.2 | | 24.4 | |
| Chen & Lee (2021) | | 59.9 | | 55.9 | | 41.8 |
| Fine-tuning | 61.8 | 67.7 | 49.9 | 62.9 | 26.5 | 48.1 |
| Subspace Reg. | **74.0** | **74.7** | **68.9** | **69.4** | 50.6 | **51.7** |
| *+language* | | | | | | |
| Cheraghian et al. (2021)* | 58.0 | | 53.0 | | 39.0 | |
| Linear Mapping | 72.6 | 73.2 | 67.1 | 68.0 | 46.9 | **50.0** |
| Semantic Subspace Reg. | 73.8 | 73.9 | 68.4 | 69.0 | 47.6 | 49.7 |
| Joint-training | | 73.1 | | 67.5 | | 47.3 |

**Feature extractor training** In session $t = 0$, we randomly select 60 classes as base classes ($|C^{(0)}| = 60$) and use the remaining 40 classes as novel classes. Reported results are averaged across 10 random splits of the data (Fig. 2). We use a ResNet-18 model that is identical to the one described in Tian et al. (2020). Following Tao et al. (2020b), we use 500 labeled samples per base class to train our feature extractor and 100 for testing.

**Incremental evaluation** Again following Tao et al. (2020b), we evaluate for a total of 8 incremental sessions $1 \leq t \leq 8$ for *mini*ImageNet. In each session, for $S^{(t)}$, we sample 5 novel classes for training and 5 samples from each class. Hence, at the last session $t = 8$, evaluation involves the entire set of 100 *mini*ImageNet classes. We use GloVe embeddings (Pennington et al., 2014) for label embeddings in Eq. 7.

**Joint-training**    In order to assess how our approaches compare to a setting where memory is abundant and retraining is allowed, we train a separate model in each session $t$ from scratch. These models are trained on the combination of all training examples including those from the base classes $\bigcup_{(t' < t)} S^{(t')}$ using the loss described in Eq. 2. Note that in this setup the memory requirement is substantially large (500 examples per each class in $C^{(0)}$, and 5 for each of the subsequent classes $C^{(1 \leq t' \leq t)}$) and the computation is more expensive.

**Results**    Fig. 2 and Table 1 show the results of multi-session experiments with and without memory. Session 0 indicates base class accuracy after feature extractor training. We compare subspace and language-guided regularization (linear mapping and semantic subspace reg.) to simple fine-tuning (a surprisingly strong baseline) and joint-training. We also compare our results to three recent benchmarks: Tao et al. (2020b), Chen & Lee (2021) and Cheraghian et al. (2021).[4]

When models are evaluated on combined base and novel accuracy, subspace regularization outperforms previous approaches (by 23% (-M) and 10% (+M) at session 8); when semantic information about labels is available, linear mapping and semantic subspace regularization outperform Cheraghian et al. (2021) (Table 1). Evaluating only base sample accuracies (Fig. 2b), semantic subspace reg. outperforms others; compared to regularization based approaches fine-tuning is subject to catastrophic forgetting. $R_{old}$ is still useful in regulating forgetting (Table 3 in appendix). The method of Chen & Lee (2021) follows a similar trajectory to our regularizers, but at a much lower accuracy (Fig. 2a). In Fig. 2c, a high degree of forgetting in base classes with fine-tuning allows higher accuracy for novel classes—though not enough to improve average performance (Fig. 2a). By contrast, subspace regularizers provide a good balance between *plasticity* and *stability* (Mermillod et al., 2013). In Table 1, storing as few as a single example per an old class substantially helps to reduce forgetting. Results from linear mapping and semantic subspace regularization are close, with semantic subspace regularization performing roughly 1% better on average. The two approaches offer different trade-offs between base and novel accuracies: the latter is more competitive for base classes and vice-versa. A similar argument can be made when comparing the subspace regularizer with the semantic counterpart where the latter is more efficient in base classes than the former. Last but not least, keeping as little as one example per old class in the memory along with our subspace regularizers is sufficient to surpass joint-training—a procedure which is extremely inefficient in terms of both space and computation.

## 5.2    SINGLE SESSION

In this section we describe the experiment setup for the single-session evaluation, $(T = 1)$, and compare our approach to state-of-the-art XtarNet (Yoon et al., 2020), as well as Ren et al. (2019), Gidaris & Komodakis (2018) and Qi et al. (2018). We evaluate our models on *1-shot* and *5-shot* settings.[5]

***mini*ImageNet and *tiered*ImageNet**    For *mini*ImageNet single-session experiments, we follow the the splits provided by Yoon et al. (2020). Out of 100, 64 classes are used in session $t = 0$, 20 in session $t = 1$ and the remaining for development. Following Yoon et al. (2020), we use ResNet-12 (a smaller version of the model described in Section 5.1). *tiered*ImageNet (Ren et al., 2018) contains a total of 608 classes out of which 351 are used in session $t = 0$ and 160 are reserved for $t = 1$. The remaining 97 are used for development. While previous work (Ren et al., 2019; Yoon et al., 2020) separate 151 classes out of the 351 for meta training, we pool all 351 for feature extractor training.

---

[4]Chen & Lee (2021) and Cheraghian et al. (2021) do not provide a codebase and Tao et al. (2020b) does not provide an implementation for the main TOPIC algorithm in their released code. Therefore, we report published results rather than a reproduction. This comparison is inexact: our feature extractor performs substantially better than Tao et al. (2020b) and Chen & Lee (2021). Despite extensive experiments (see appendix) on various versions of ResNet-18 (He et al., 2016), we were unable to identify a training procedure that reproduced the reported accuracy for session 0: all model variants investigated achieved 80%+ validation accuracy.

[5]Unlike in the preceding section, we were able to successfully reproduce the XtarNet model. Our version gives better results on the *mini*ImageNet dataset but worse results on the *tiered*ImageNet datasets; for fairness, we thus report results for *our version* of XtarNet on *mini*ImageNet and *previously reported* numbers on *tiered*ImageNet. For other models, we show accuracies reported in previous work.

Table 2: *mini*ImageNet 64+5-way and *tiered*ImageNet 200+5-way single-session results. We follow previous work in reporting the average of accuracies of base and novel samples over all classes rather than weighted average. In addition to accuracy, we report a quantity labeled $\Delta$ by Ren et al. (2019), which is the gap between individual accuracies and joint accuracies of both base and novel samples averaged. Lower values of $\Delta$ are better. Bold numbers are not significantly different from the best result in each column under a paired t-test ($p < 0.05$ after Bonferroni correction). All results are averaged across 2000 runs.

| | *mini*ImageNet | | | | *tiered*ImageNet | | | |
|---|---|---|---|---|---|---|---|---|
| | 1-shot | | 5-shot | | 1-shot | | 5-shot | |
| Model | Acc. | $\Delta$ | Acc. | $\Delta$ | Acc. | $\Delta$ | Acc. | $\Delta$ |
| Imprinted Networks (Qi et al., 2018) | $41.34_{\pm 0.54}$ | -23.79% | $46.34_{\pm 0.54}$ | -25.25% | $40.83_{\pm 0.45}$ | -22.29% | $53.87_{\pm 0.48}$ | -17.18% |
| LwoF (Gidaris & Komodakis, 2018) | $49.65_{\pm 0.64}$ | -14.47% | $59.66_{\pm 0.55}$ | -12.35% | $53.42_{\pm 0.56}$ | -9.59% | $63.22_{\pm 0.52}$ | -7.27% |
| Attention Attractor Networks (Ren et al., 2019) | $54.95_{\pm 0.30}$ | -11.84% | $63.04_{\pm 0.30}$ | -10.66% | $56.11_{\pm 0.33}$ | -6.11% | $65.52_{\pm 0.31}$ | -4.48% |
| XtarNet (Yoon et al., 2020) | $56.12_{\pm 0.17}$ | -13.62% | $69.51_{\pm 0.15}$ | -9.76% | $61.37_{\pm 0.36}$ | -1.85% | $69.58_{\pm 0.32}$ | -1.79% |
| Fine-tuning | $58.56_{\pm 0.33}$ | -12.14% | $66.54_{\pm 0.33}$ | -13.77% | $\mathbf{64.42}_{\pm 0.38}$ | -7.23% | $72.59_{\pm 0.34}$ | -6.88% |
| Subspace Regularization | $58.38_{\pm 0.32}$ | -12.30% | $68.88_{\pm 0.32}$ | -10.74% | $\mathbf{64.39}_{\pm 0.38}$ | -7.23% | $\mathbf{73.03}_{\pm 0.34}$ | -6.16% |
| *+language* | | | | | | | | |
|   Linear Mapping | $\mathbf{58.87}_{\pm 0.33}$ | -12.83% | $\mathbf{69.68}_{\pm 0.31}$ | -10.40% | $\mathbf{64.55}_{\pm 0.38}$ | -7.31% | $\mathbf{73.10}_{\pm 0.33}$ | -6.16% |
|   Semantic Subspace Reg. (w/ description) | $\mathbf{59.09}_{\pm 0.32}$ | -12.38% | $68.46_{\pm 0.32}$ | -11.70% | $\mathbf{64.49}_{\pm 0.38}$ | -7.14% | $72.94_{\pm 0.34}$ | -6.29% |
|   Semantic Subspace Reg. (w/ label) | $58.70_{\pm 0.32}$ | -12.24% | $\mathbf{69.75}_{\pm 0.32}$ | -10.48% | $\mathbf{64.75}_{\pm 0.38}$ | -7.22% | $\mathbf{73.51}_{\pm 0.33}$ | -6.08% |

We train the same ResNet-18 described in Section 5.1. Additional details regarding learning rate scheduling, optimizer parameters and other training configurations may be found in the appendix.

**Incremental evaluation** We follow Yoon et al. (2020) for evaluation. Yoon et al. independently sample 2000 $D^{(1)}$ incremental datasets ("episodes") from the testing classes $C^{(1)}$. They report average accuracies over all episodes with 95% confidence intervals. At every episode, $Q^{(1)}$ is resampled from both base and novel classes, $C^{(0)}$ and $C^{(1)}$, with equal probability for both *mini*ImageNet and *tiered*ImageNet. We again fine-tune the weights until convergence. We do not reserve samples from base classes, thus the only training samples during incremental evaluation is from the novel classes $C^{(1)}$. We use the same resources for label embeddings and Sentence-BERT embeddings (Reimers & Gurevych, 2019) for descriptions which are retrieved from WordNet (Miller, 1995).

**Results** We report aggregate results for 1-shot and 5-shots settings of *mini*ImageNet and *tiered*ImageNet (Table 2). Compared to previous work specialized for the single-session setup without a straightforward way to expand into *multi-session* (Ren et al., 2019; Yoon et al., 2020), even our simple fine-tuning baseline perform well on both datasets—outperforming the previous state-of-the-art in three out of four settings in Table 2. Addition of subspace and semantic regularization improves performance overall but *tiered*ImageNet 1-shot setting. Semantic subspace regularizers match or outperform linear label mapping. Subspace regularization outperforms fine-tuning in 5-shot settings and matches it in 1-shot. In addition to accuracy, we report a quantity labeled $\Delta$ by Ren et al. (2019). $\Delta$ serves as a measure of catastrophic forgetting, with the caveat that it can be minimized by a model that achieves a classification accuracy of 0 on both base and novel classes. We find that our approaches result in approximately the same $\Delta$ in *mini*ImageNet and worse in *tiered*ImageNet than previous work.

## 6 ANALYSIS AND LIMITATIONS

**What does regularization actually do?** Fine-tuning results in prediction biased towards the most recently learned classes (top of Fig. 3) when no subspace regularization is imposed. Our experiments show that preserving the base weights while regularizing novel weights gives significant improvements over ordinary fine-tuning (bottom of Fig. 3)—resulting a fairer prediction over all classes and reducing catastrophic forgetting. In Table 1, Semantic Subspace Reg. results in 73.8% and 47.6% accuracies in the 1[st] and 8[th] sessions whereas, fine-tuning results in 61.8% and 26.5%, respectively, even without any memory—suggesting that regularization ensures a better retention of accuracy. While the trade-off between accuracies of base and novel classes is inevitable due to the nature of the classification problem, the proposed regularizers provide a good balance between the two.

**What are the limitations of the proposed regularization scheme?** Our approach targets only errors that originate in the final layer of the model—while a convolutional feature extractor is used, the parameters of this feature extractor are fixed, and we have focused on FSCIL as a linear clas-

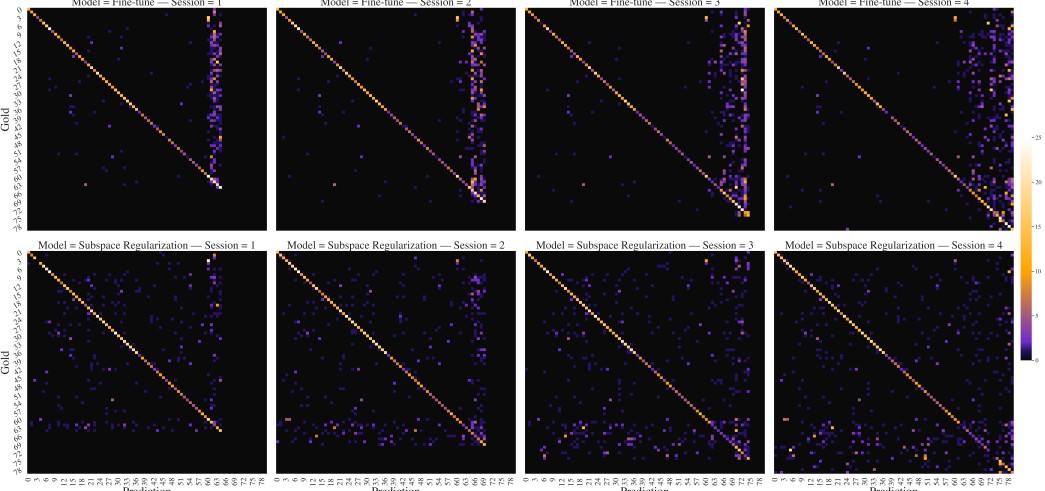

Figure 3: Simple fine-tuning (top) vs. subspace regularization (bottom) predictions without memory across the first *four* incremental sessions of *mini*ImageNet. In the *x*- and *y*-axes, we present predictions and gold labels ranging from 0 to 79 where the first 60 are base classes. The number of classes grows by 5 every session starting from 60 up to 80. Brighter colors indicate more frequent predictions. Note that simple fune-tuning entails bias towards the most recently learned classes (top row) whereas addition of subspace regularization on the novel weights remedies the aforementioned bias; resulting in a fairer prediction performance for all classes.

sification problem. Future work might extend these approaches to incorporate fine-tuning of the (nonlinear) feature extractor itself while preserving performance on all classes in the longer term.

# 7 CONCLUSIONS

We have described a family of regularization-based approaches to few-shot class-incremental learning, drawing connections between incremental learning and the general multi-task and zero-shot learning literature. The proposed regularizers are extremely simple—they involve only one extra hyperparameter, require no additional training steps or model parameters, and are easy to understand and implement. Despite this simplicity, our approach enables ordinary classification architectures to achieve state-of-the-art results on the doubly challenging *few-shot incremental* image classification across multiple datasets and problem formulations.

## ACKNOWLEDGMENTS

This work is supported in part by the U.S. NSF grant 1838193 and DARPA HR001118S0044 (the LwLL program). The U.S. Government is authorized to reproduce and distribute reprints for Governmental purposes. The views and conclusions contained in this publication are those of the authors and should not be interpreted as representing official policies or endorsements of DARPA and the U.S. Government. Computing resources were provided by the MIT Supercloud (Reuther et al., 2018).

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

## A  CODE AND DATASETS

Code is released under https://github.com/feyzaakyurek/subspace-reg. We use *mini*ImageNet (Vinyals et al., 2016) and *tiered*ImageNet (Ren et al., 2018) datasets both are subsets of ImageNet dataset (Russakovsky et al., 2015). Use of terms and licenses are available through the respective sources.

## B  ANALYSIS OF CATASTROPHIC FORGETTING

In Table 3, we demonstrate the effectiveness of the regularization term $R_{old}$ in mitigating catastrophic forgetting.

Table 3: *mini*ImageNet weighted average results across multiple sessions showcasing the usefulness of $R_{old}$ in reducing catastrophic forgetting across multiple sessions for -M setting. Higher accuracies are highlighted and results are averages over 10 random splits. $R_{old}$ is useful in combination with Subspace Reg. for all sessions while it is more helpful in the long-run for Fine-tuning. Note that Subspace Reg. consistently outperforms Fine-tuning regardless the use of $R_{old}$.

| Model | 0 | 1 | 2 | 3 | 4 | 5 | 6 | 7 | 8 |
|---|---|---|---|---|---|---|---|---|---|
| Subspace Reg. | 80.37 | **72.99** | **67.21** | **62.40** | **58.46** | **54.28** | **50.33** | **48.03** | **45.06** |
| Subspace Reg. no $R_{old}$ | 80.37 | 71.96 | 65.49 | 60.43 | 55.56 | 50.35 | 45.66 | 42.91 | 39.54 |
| Fine-tuning | 80.37 | 61.77 | 49.93 | 40.45 | 34.04 | 31.63 | **28.43** | **27.91** | **26.54** |
| Fine-tuning no $R_{old}$ | 80.37 | **62.39** | **53.89** | **46.56** | **39.73** | **32.92** | 27.00 | 23.95 | 20.39 |

Table 4: *mini*ImageNet +M results across multiple sessions in tabular form. Initial number of base classes is 60 and 5 new classes are introduced at every session. Results are on the test set that grows with the increasing number of classes. In the last session we evaluate over all 100 classes. Joint-training involves keeping all training data seen thus far in the memory and retrains a feature extractor from scratch on the entirety of base and novel classes.

| Session
Model | 0 | 1 | 2 | 3 | 4 | 5 | 6 | 7 | 8 |
|---|---|---|---|---|---|---|---|---|---|
| Chen & Lee (2021) | 64.77 | 59.87 | 55.93 | 52.62 | 49.88 | 47.55 | 44.83 | 43.14 | 41.84 |
| Fine-tuning | 80.37 | 67.69 | 62.91 | 59.52 | 56.87 | 54.37 | 51.92 | 50.26 | 48.13 |
| Subspace Reg. | 80.37 | 74.68 | 69.39 | 65.51 | 62.38 | 59.03 | 56.36 | 53.95 | 51.73 |
| *+language* | | | | | | | | | |
|   Linear Mapping | 80.37 | 73.24 | 67.96 | 64.50 | 61.28 | 57.68 | 54.64 | 52.25 | 50.00 |
|   Semantic Subspace Reg. | 80.37 | 73.92 | 69.00 | 65.10 | 61.73 | 58.12 | 54.98 | 52.21 | 49.65 |
| Joint-training | 80.37 | 73.06 | 67.52 | 63.21 | 59.19 | 55.98 | 52.11 | 49.59 | 47.31 |

Table 5: *mini*ImageNet -M results across multiple sessions in tabular form. Initial number of base classes is 60 and 5 new classes are introduced at every session. Results are on the test set that grows with the increasing number of classes. The last session is evaluated over all 100 classes. *Note that the entries for Cheraghian et al. (2021) are only rough estimates from the visual plot provided in their published work.

| Model | 0 | 1 | 2 | 3 | 4 | 5 | 6 | 7 | 8 |
|---|---|---|---|---|---|---|---|---|---|
| Tao et al. (2020) | 61.31 | 50.09 | 45.17 | 41.16 | 37.48 | 35.52 | 32.19 | 29.46 | 24.42 |
| Fine-tuning | 80.37 | 61.77 | 49.93 | 40.45 | 34.04 | 31.63 | 28.43 | 27.91 | 26.54 |
| Subspace Regularization | 80.37 | 74.01 | 68.92 | 64.37 | 61.35 | 58.22 | 54.99 | 53.15 | 50.58 |
| *+language* | | | | | | | | | |
|   Cheraghian et al. (2021)* | 62.00 | 58.00 | 52.00 | 49.00 | 48.00 | 45.00 | 42.00 | 40.00 | 39.00 |
|   Linear Mapping | 80.37 | 72.65 | 67.11 | 63.47 | 59.82 | 55.44 | 51.42 | 49.64 | 46.90 |
|   Semantic Subspace Reg. | 80.37 | 73.76 | 68.36 | 64.07 | 60.36 | 56.27 | 53.10 | 50.45 | 47.55 |

## C  RESULTS IN TABULAR FORM

In Table 4 and Table 5, we present the multi-session results in the main paper in the tabular form.

## D  DETAILS OF FEATURE EXTRACTOR TRAINING

We use the exact ResNet described in Tian et al. (2020), the differences compared to the standard ResNet (He et al., 2016): (1) Each block (collection of convolutional blocks) is composed of three convolutional layers instead of two. (2) Number of blocks for ResNet-12 is 4 instead of 6 of the standard version, thus the total number of convolutional layers are the same. (3) Filter sizes are [64,160,320,640] rather than [64,128,256,512], though the total number of filters is comparable since Tian et al. (2020) has less blocks. (4) There is Dropblock at the end of the last blocks.

Tian et al. (2020) provides a full visualization in Appendix and their code repository[6] is easy to browse on which we base our own codebase. We observe that the previous work oftentimes use their slightly modified version of the standard ResNet. Ren et al. (2019) uses the ResNet-10 (Mishra et al., 2017) and ResNet-18 for for *mini*ImageNet and *tiered*ImageNet, respectively. XtarNet(Yoon et al., 2020) is originally based on a slightly modified version of ResNet-12 and ResNet-18 which we replaced with our version, improving their results for *mini*ImageNet but not in *tiered*ImageNet, thus we report improved results for *mini*ImageNet and their results for *tiered*ImageNet in the main paper.

---

[6] https://github.com/WangYueFt/rfs

Table 6: *mini*ImageNet validation set accuracy with two ResNet-18 architectures with slight differences as listed in Appendix D. Overall performances are comparable.

|  | Seed 1 | Seed 2 | Seed 3 | Seed 4 | Seed 5 | Seed 6 | Seed 7 | Seed 8 | Seed 9 | Seed 10 | Mean |
|---|---|---|---|---|---|---|---|---|---|---|---|
| Our ResNet-18 (Tian et al., 2020) | 84.833 | 79.167 | 83.200 | 81.300 | 81.267 | 78.933 | 82.033 | 82.067 | 81.800 | 82.367 | 81.6967 |
| Standard ResNet-18 (He et al., 2016) | 83.333 | 80.100 | 83.867 | 81.333 | 80.967 | 79.100 | 81.833 | 82.500 | 81.167 | 81.567 | 81.5767 |

### D.1 DEFAULT SETTINGS

Unless otherwise indicated we use the following default settings of Tian et al. (2020) in our feature extractor training. We use SGD optimizer with learning starting at 0.05 with decays by 0.1 at epochs 60 and 80. We train for a total of 100 epochs. Weight decay is 5e-4, momentum is 0.9 and batch size is 64. As per transformations on training images, we use random crop of 84x84 with padding 8. We also use color jitter (*brightness*=0.4, *contrast*=0.4, *saturation*=0.4) and horizontal flips. For each run, we sample 1000 images from base classes and 25 images from each of novel classes. Our classifier does not have bias.

### D.2 MULTI-SESSION *mini*IMAGENET

We re-sample the set of base classes (60 classes) 10 times across different seeds and train ten ResNet-18 models. Each class has 500 training images. We follow the default settings for training.

### D.3 MULTI-SESSION COMPARISON TO STANDARD RESNET-18

In Table 6 we provide validation set results for two types of ResNet-18's: Tian et al. (2020) and He et al. (2016) across ten different seeds. Results show that use of Tian et al. (2020) does not incur unfair advantage over those who used He et al. (2016).

### D.4 SINGLE-SESSION *mini*IMAGENET

We follow the default hyperparameters parameters Appendix D.1 except that for training, validation and testing we use the exact splits provided by Ren et al. (2019) also used by Yoon et al. (2020). There are 64 base, 16 validation and 20 testing classes provided (totaling 100). Training data consists of 600 images per base class. Dataset statistics are delineated in the Appendix of Ren et al. (2019) and downloadable splits are available here, courtesy of Ren et al. (2019).

### D.5 SINGLE-SESSION *tiered*IMAGENET

*tiered*ImageNet is first introduced by Ren et al. (2018). Same as above, we use the default parameters except that we train for a total of 60 epochs decaying the initial learning rate of 0.05 by 0.1 at epochs 30 and 45. Again, we use the same data as previous work available at the same link above. *tiered*ImageNet is split into 351, 97 and 160 classes. Past work that use meta-learning Ren et al. (2019); Yoon et al. (2020) split 351 training classes into further 200 and 151 clases where the latter is used for meta learning. We pool all 351 for feature extractor training. At the end of feature extractor training, we only keep the classifier weights for the first 200 classes to adhere to the evaluation scheme of 200+5 classes as past work.

## E DETAILS OF INCREMENTAL EVALUATION

### E.1 QR DECOMPOSITION FOR SUBSPACE REGULARIZATION

To compute the orthogonal basis $P_{C^{(0)}}$ for the subspace spanned by base classifier weights $\eta_{C^{(0)}}$ we use QR decomposition(Trefethen & Bau III, 1997):

$$\begin{bmatrix} P_{C^{(0)}} & Q' \end{bmatrix} \begin{bmatrix} R \\ \mathbf{0} \end{bmatrix} = \eta_{C^{(0)}}^{\top} \tag{11}$$

### E.2 MULTI-SESSION

For testing, we sample 1000 images from base classes and 25 images from each of novel classes. Testing images from a given class stay the same across sessions. Harmonic mean results take into account the ratio of base classes to novel classes in a given session. In this setting, there is no explicit development set (with disjoint classes than train and test) defined by previous work thus we use the first incremental learning session (containing 5 novel classes) as our development set.

**Default settings**   We use the same transformations as in Appendix D.1 on the training images. We stop fine-tuning when loss does not change more than 0.0001 for at least 10 epochs. We use SGD optimizer. We repeat the experiments 10 times and report the average accuracy with standard deviation (95% confidence interval) in the paper.

**Simple Fine-tuning**   We use learning rate of 0.002 and do not use learning decay. Weight-decay $\alpha$ is set at 5e-3. In order to limit the change in weights, we use different $\beta$'s for base and previously learned novel classes, where the former is 0.2 and the latter 0.1. We rely on the default settings otherwise.

**Subspace Regularization**   Different from simple fine-tuning we use a weight decay of 5e-4. There is an additional parameter called $\gamma$ in this setting controlling the degree of pulling of novel weights toward the subspace which we set to 1.0.

**Semantic Subspace Regularization**   Different than simple subspace regularization, there is a temperature parameter used in the Softmax operation used in computation of $l_c$'s which we set to 3.0.

**Linear mapping regularization**   Same parameters as in subspace regularization are used except $\gamma = 0.1$. We formulate $L$ as a linear layer with bias and we use gradient descent to train the parameters.

### E.3 SINGLE-SESSION

In our 1-shot experiments unless fine-tuning converges by then, we stop at the maximum number of epochs at 1000. We sample 2000 episodes which includes 5 novels classes and 1-5 samples from each and report average accuracy. For testing, base and novel samples have equal weight in average per previous work (Yoon et al., 2020). SGD optimizer is used. For description similarity we use Sentence-BERT's `stsb-roberta-large`.

***mini*ImageNet Settings**   We use the same set of transformations on the training images as described in Appendix D.1. We first describe details of 1-shot setting. In 1-shot experiments we set the maximum epochs to 1000. For simple fine-tuning we use learning rate of 0.003, and weight decay of 5e-3. In Semantic Subspace Reg., we set temperature to 1.5. Both in Semantic Subspace Reg. and linear mapping $\gamma = 0.005$ and weight-decay is 5e-4. In subspace regularization, $\gamma = 0.005$ and weight-decay is set to 5e-5. Description similarity follows the same setup as Semantic Subspace Reg..

In 5-shot setting, we set $\beta = 0.03$ weight-decay to 5e-3 and learning rate to 0.002. For subspace regularization, Semantic Subspace Reg. and linear mapping we use $\gamma = 0.03$ and for description similarity we use $\gamma = 0.01$.

***tiered*ImageNet Settings**   In 1-shot setting, fine-tuning uses learning rate of 0.003. Semantic Subspace Reg. has learning rate of 0.005, weight-decay 5e-3 and $\gamma = 0.005$. Subspace reg. and linear mapping use $\gamma = 0.001$.

In 5-shot setting, for simple fine-tuning we set lr = 0.001, weight-decay=5e-3, $\beta = 0.3$. For Semantic Subspace Reg. $\gamma = 0.05$ and $\beta = 0.2$ while others have $\gamma = 0.03$.

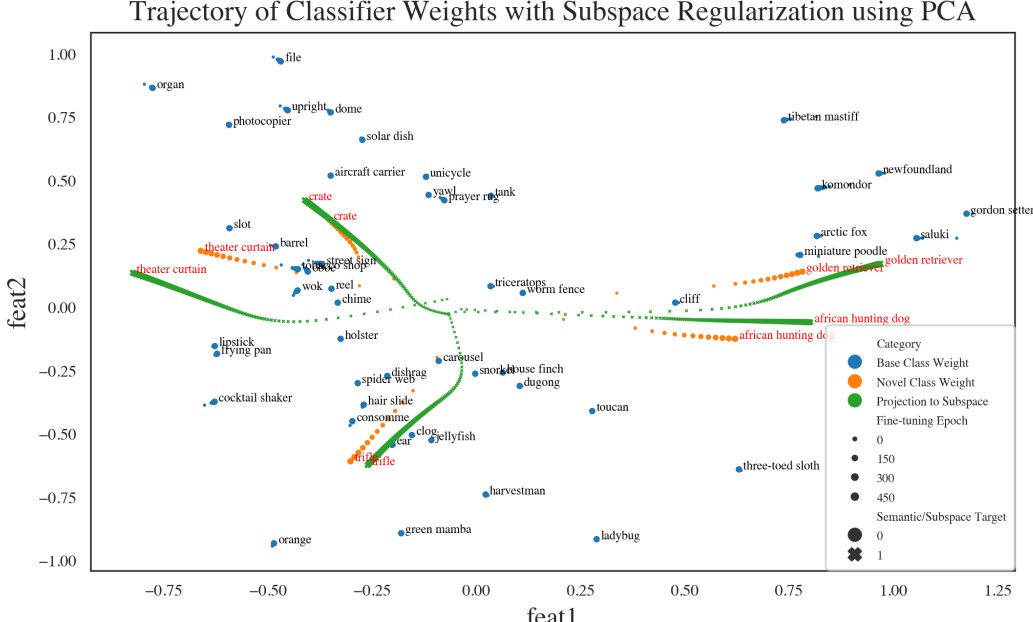

Figure 4: Clasifier weight space when subspace regularization is applied for *mini*ImageNet single-session 5-shot setting. First two principal components are shown according to PCA. Red labels indicate novel classes while the black indicates base. The green crosses indicate the projection of the respective novel class weight to the base subspace. Note that unlike label/description similarity and linear mapping, *subspace target* is dynamic: it changes according to its corresponding novel weights and vice versa.

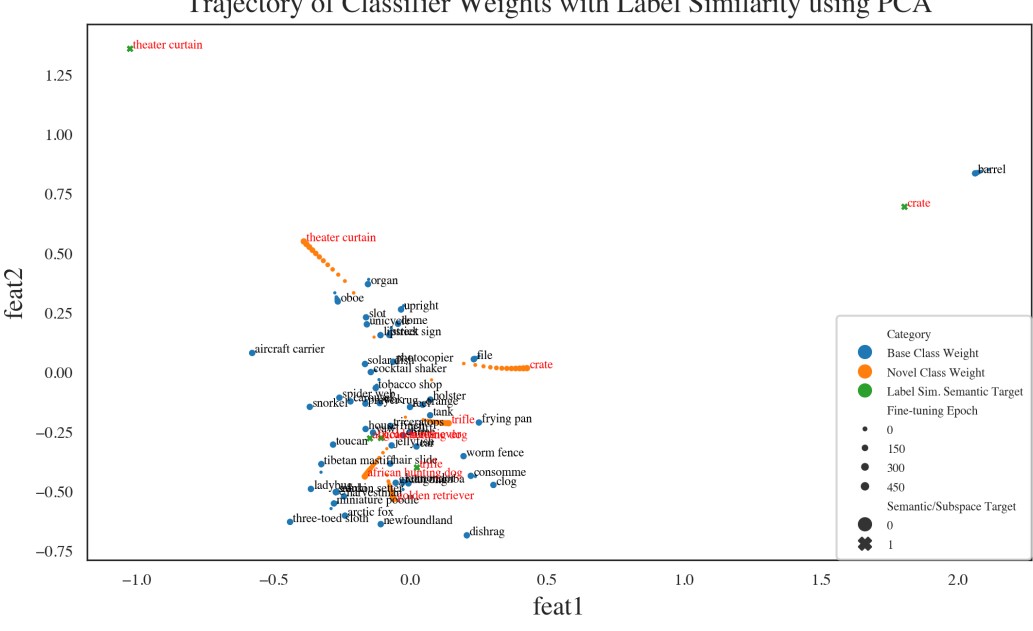

Figure 5: Clasifier weight space when Semantic Subspace Reg. is applied for *mini*ImageNet single-session 5-shot setting. First two principal components are shown according to PCA. Red labels indicate novel classes while the black indicates base. The green crosses indicate the *semantic target l* of the respective novel class. Note that semantic targets are static: they don't change during fine-tuning. Notably, the semantic target for *theater curtain* falls closely to the class representation of the base class *stage*, dragging novel weight for theater curtain towards there. Same dynamic is visible for novel class *crate* and base *barrel*.

## F   VISUALIZATIONS

In Fig. 4 and Fig. 5 we depict principal components of classifier weights as well as semantic or subspace targets for novel weights.

## G   COMPUTE

We use a single 32 GB V100 NVIDIA GPU for all our experiments.

