# OpenReview forum: "Subspace Regularizers for Few-Shot Class Incremental Learning"
_ICLR.cc/2022/Conference — ICLR 2022 Poster_

### Official Review · Reviewer_oQAd · 2021-10-27

**Correctness:** 3
**Technical Novelty And Significance:** 3
**Empirical Novelty And Significance:** 3
**Recommendation:** 6
**Confidence:** 5

**Main Review:**

Strengths
=======
* The method is technically sound, simple, and it improves the performance on multi-session FSCIL.
* The text is well-written.
* The authors provide enough details to reproduce their results and they promise to release the code public.

Weaknesses
==========
* The improvement on the single-session setup is not clear. However, I value that the authors were honest about it. In fact, in this setup simple fine-tuning seems to be very effective.
* There are some relevant works that were not cited such as [A,B,C,D,E].
* Some previous work such as [C, Cheraghian et al.] evaluated their models on CUB and CIFAR. Could you clarify why you did not use the same evaluation as them?
* The main contribution (aligning new weights with previously learned ones) is not new (the authors already cite previous work), but the way they do it seems new to me. It would be good if you clarified what exactly in your work is proposed by you and what is borrowed from previous literature.

[A] Simon, Christian, et al. "Adaptive subspaces for few-shot learning." Proceedings of the IEEE/CVF Conference on Computer Vision and Pattern Recognition. 2020.

[B] Tao, Xiaoyu, et al. "Topology-preserving class-incremental learning." European Conference on Computer Vision. Springer, Cham, 2020.

[C] Tao, Xiaoyu, et al. "Few-shot class-incremental learning." Proceedings of the IEEE/CVF Conference on Computer Vision and Pattern Recognition. 2020.

[D] (Arxiv, Optional) Yan, Kun, et al. "Aligning Visual Prototypes with BERT Embeddings for Few-Shot Learning." arXiv preprint arXiv:2105.10195 (2021).

[E] Xing, Chen, et al. "Adaptive cross-modal few-shot learning." Advances in Neural Information Processing Systems 32 (2019): 4847-4857.


**Summary Of The Paper:**

The authors tackle the problem of few-shot class incremental learning (FSCIL). In this setting, a learner is first trained on a set of base classes for which plenty of data is available. Then, it must learn from small sets of novel classes while retaining good performance in the whole sequence. In order to prevent the classifier from overfitting new classes, they authors propose to regularize novel class weights to keep them in the subspace of base classes, thus making the classifier rely on known features rather than new features that are probably spurious. In addition, they also show that it is possible to align new classes with base classes based on their similarity in a learned semantic space such as Glove. Experimental results on miniImagenet and tiered-Imagenet show that their method obtains better performance than previous state of the art.

**Summary Of The Review:**

The proposed method is simple and effective and the paper is well-written. On the other hand, the main contribution is small and I have some questions about the experimental setup and relationship with previous related work (see weaknesses). Overall, I think that the simplicity and effectiveness outweigh the negative aspects and **I would be inclined towards accept if the authors resolve my concerns and those of the other reviewers.**

---

> ### Author Response · Authors · 2021-11-16
> **Response to Reviewer oQAd**
>
> Thank you very much for the thorough review! We hope your concerns are addressed by the discussion below; please let us know if there is any information we can provide.
>
> **Re1:** “Improvements in single-session setup.”
>
> We agree that simple fine-tuning is already a strong baseline---indeed, we view the success of this baseline as one of the contributions of our work to the few-shot / incremental learning literature! However, over 2000 novel episodes, semantic subspace regularization consistently improves upon fine-tuning by 1-3% points under the single-session 5-shot settings and better preserves base class performance in the multi-session setting.
>
> **Re2:** “Missing references A, B, C, D and E.”
>
> Thank you, references are added. For comparison to Simon et al. (2020) kindly refer to our response Re2 to Reviewer Vrap. Is [C] different from Tao et al. (2020) in Table 1? Xing et al. [E] is similar to our work in using semantic information to shape the prototype space; however, they do not exploit subspace information of the feature space and do not explore performance in an incremental setup.
>
> **Re3:** “Evaluating on CUB and CIFAR100.”
>
> After strenuous yet failed attempts to replicate previous work (Chen et al., Tao et al., ) for CUB, we have contacted the respective authors and were told that they both start with a pretrained backbone on ImageNet, whose training set contains examples of novel classes! On the [website](http://www.vision.caltech.edu/visipedia/CUB-200.html) for CUB200, below warning is provided:
>
> _“Warning: Images in this dataset overlap with images in ImageNet. Exercise caution when using networks pretrained with ImageNet (or any network pretrained with images from Flickr) as the test set of CUB may overlap with the training set of the original network.”_
>
> Thus, in keeping with past work, we focus on the cleaner evaluation provided by tiered/miniImageNet.
>
> **Re4:** “Comparison to previous attempts on aligning old and new weights.”
>
> Thank you! At the end of Section 4.3 we already provide comparisons to Argyriou et al. (2017) and Agarwal et al. (2010). In the revision, we have augmented that section to include some other related works in subspace regularization. While we provide individual comparisons to relevant references below, the summary is:
>
> (1) these works do not target incremental learning and cannot be straightforwardly extended to accommodate it.
>
> (2) some of these approaches require test-time access to privileged information about the identity of the example to be classified.
>
> (3) to the best of our knowledge, the specific subspace regularizer in Eq. 5 in our paper, despite its simplicity, has not previously been used in this form.
>
> **Argyriou et al. (2007):** This work is similar to our work based on their intuition that related tasks could benefit from a common set of features. In our work, we estimate these features to be the ones learned during pre-training. In addition to (3) above their work is different in that they simultaneously train on multiple tasks together requiring all training data to be available at once.
>
> **Jacob et al. (2008):** Adopts a similar setup to Evgeniou & Pontil (2007). In their work they show that assuming an unknown cluster prior over all tasks where similar tasks are grouped together and weights should be similar within a group is useful in multi-task learning. They do not explicitly enforce a subspace constraint. While the intuition that similar tasks within a group should have similar weights is related to our semantic subspace regularizer, their problem setup is substantially different than ours.
>
> **Agarwal et al. (2010):** This work is closest to ours as it models all task parameters on the same manifold and enforces this constraint through regularization. It is different as they learn the manifold and the task parameters alternately and require all task data to be available before hand (1).

---

> > ### Comment · Reviewer_oQAd · 2021-11-19
> > **Response to authors**
> >
> > Thanks for your response. After reading your response to my comment, the response to the other reviewers, and the revised version of the paper, I am inclined towards raising my score.

---

> > > ### Author Response · Authors · 2021-11-29
> > > **Thanks**
> > >
> > > Thank you for your positive recommendation!

---

### Official Review · Reviewer_Vrap · 2021-11-02

**Correctness:** 4
**Technical Novelty And Significance:** 3
**Empirical Novelty And Significance:** 3
**Recommendation:** 6
**Confidence:** 4

**Main Review:**

Strengths:
- The paper presents a simple model and strong experimental results.
- The experimentation seems solid. It has evaluated on single and multi-session setups, with or without a memory buffer.

Weaknesses:
- It is unclear where the most benefit of the proposed model comes from. Based on Figure 2, it seems that Chen & Lee (2021) has a much worse checkpoint for base classes, and perhaps as a result the novel class learning is also not as good. It would be good if the authors can experiment both Tao et al. (2020) and Chen & Lee (2021) with the same pretrained checkpoint to showcase the real benefit in the few-shot learning phase.
- The type of subspace regularization exists in non-incremental few-shot learning and I wonder if the authors can provide an empirical comparison on that. It is unclear why subspace regularization depends on the incremental aspect. See Simon et al. (2020), Devos & Grossglauser (2020), Yoon et al. (2019).
- The semantic regularization technique provides a small gain on the multi-session benchmark and very marginal gain on the single session benchmark. This makes the contribution less focused and I suggest only include this in an additional study in the experiment section.
- As pointed out by the authors, the current framework only aims at learning the linear classifier in a sequential manner, which limits the ways it can be applied.
- If the number of samples is small (as is the case for few-shot learning sessions), then one could afford to do joint training at each session and store all the support examples. Such a joint training baseline should be provided for comparison (as was provided in Chen & Lee).

References:
- Simon et al. Adaptive Subspaces for Few-Shot Learning. CVPR 2020.
- Devos and Grossglauser. Regression Networks for Meta-Learning Few-Shot Classification. ICML Workshop on Automated Machine Learning 2020.
- Yoon et al. TAPNet: Neural Network Augmented with Task-Adaptive Projection for Few-Shot Learning. ICML 2019.


**Summary Of The Paper:**

The paper proposes subspace regularization technique for incremental few-shot learning. The high-level idea is to find the basis vectors of the subspace spanned by the base classes, and then project the new classes into the subspace. The regularizer encourages the new class weights to be similar to the projected vectors with an L2 loss. Additionally, the paper also proposes to use semantic word vectors to perform regularization. Experimental results confirm that the proposed method is significantly better compared to Tao et al. (2020) and Chen & Lee (2021) either with an additional memory buffer or without, but it is unclear whether it is due to a better few-shot learning algorithm or simply a better pretrained checkpoint on the base classes. Moreover, the benefit of semantic regularization is not significant.

**Summary Of The Review:**

In conclusion, my main concern is that this work may not have the same starting checkpoint when comparing to prior works and therefore it is hard to evaluate its empirical success in the few-shot learning phase. Another main concern is that the relation between subspace and incremental learning is unclear, since subspace regularization could also be applied to standard few-shot learning, and therefore comparisons to subspace methods on standard few-shot learning seems required. Other aspects of this work look solid to me. My initial evaluation of this paper is “weak reject”.

Authors' response on joint training addressed my concern. However, my other concern about comparison to Tao 2020 and Chen & Lee 2021 was not fully addressed. That being said, I understand there might be difficulty reproducing the numbers. Overall, I increased my rating from 5 to 6 to acknowledge the effort in responding my review comments.

---

> ### Author Response · Authors · 2021-11-16
> **Response to Reviewer Vrap**
>
> Thank you for the thoughtful review! Please let us know if you have any additional questions (especially about the relationship to past work) after the discussion below:
>
> **Re1:** “Replicating Tao el al. (2020) and Chen and Lee. (2021).”
>
> Tao et al. (2020) does not provide the main TOPIC algorithm in the released codebase ([link to github issue](https://github.com/xyutao/fscil/issues/25)) and have been unresponsive through email. We have tried downloading their pretrained backbone however, the download link does not work ([link to the issue](https://github.com/xyutao/fscil/issues/21)). Chen and Lee does not provide a codebase so we have re-implemented it based on the paper. Their cross-entropy loss uses an exponential kernel which takes in a norm term defined over a high-dimensional space (Eq. 1 in the paper) involving a set of trainable weights called reference vectors and $\gamma$. Their initializations are critical in preventing gradients from winding down to zero. The accuracy on the base set from our best attempt to implement the kernel method described in the paper is 15% suggesting that some important piece of information is clearly missing.
>
> Reviewer qioC brought concurrent work [1] to our attention which also reports 80+% accuracy on base classes. From what we can read in Fig 5b, their model’s performance is within 1% of our approach, which is considerably simpler and more parameter-efficient than [1].
>
> In addition to these best-effort comparisons, we hope that questions about the relation of our results to past work are addressed by comparisons to XtarNet in other sections of the paper.
>
>
> **Re2:** “Subspace-based classification in few-shot learning.”
>
> Thanks for pointing out this related work. It is referenced in the revision in Sections 2 and 4.3. As you have noted, these works all focus on the non-incremental FSL setting where the number of classes to choose from is fixed which lacks the most central challenge in incremental learning i.e. joint prediction. For more detailed discussion of comparisons to related work:
>
> **Simon et al. (2020) and Devos and Grossglauser (2020):** They define subspaces (orthogonal in Devos et al.) for every class and predict based on the distances to each of these subspaces. Instead we learn new parameters for novel classes through regularized logistic regression where the regularization is based on a soft-projection onto the base subspace. That is, rather than using disjoint subspaces, we encourage a joint subspace. While a modification may be possible, their method is not applicable to incremental learning as is.
>
> **Yoon et al. (2019):** Yoon et al. learn to do task-specific conditioning by projecting features linearly to some other subspace where they are more distinguishable. In fact, XtarNet [2] by the same authors provides an extension of TARs into the single-session incremental learning where the authors introduce a meta-learning stage during which they learn how to extract features from the novel examples and also how to combine them with older features. While this is an intriguing future direction to explore, how should the meta learning stage be modified to incorporate many batches of novel classes is not straightforward.
>
> **Re3:** “Minor gain from semantic regularization.”
>
> Thank you for your suggestions in restructuring the paper, we will take them into account in writing the final version.
>
> **Re4:** “Providing a joint training baseline.”
>
> Below we provide results for a joint training where at session $t$ we train ResNet-18 from scratch over the union of $\{D^{(0)}, D^{(1)},\dots, D^{(t)} \}$. Note that in this baseline, memory size effectively is as large as the base training set which is 30,000 images for miniImageNet and grows in every session. In the below table, our semantic subspace regularizer with no memory is within 1% difference with and consistently above joint-training.
>
> |Model|   0 |    1 |    2 |    3 |    4 |    5 |    6 |    7 |    8 |
> |:-----------------------|--------:|-----:|-----:|-----:|-----:|-----:|-----:|-----:|-----:|
> | Fine-tune              |    80.4 | 61.8 | 49.9 | 40.5 | 34.0   | 31.6 | 28.4 | 27.9 | 26.5 |
> | Subspace Reg.          |    80.4 | 71.7 | 66.9 | 62.5 | 58.9 | 55.0   | 51.9 | 49.8 | 46.8 |
> | Semantic Subspace Reg. |    80.4 | 73.8 | 68.4 | 64.1 | 60.4 | 56.3 | 53.1 | 50.5 | 47.6 |
> | Joint-training         |    80.4 | 73.1 | 67.5 | 63.2 | 59.2 | 56.0   | 52.1 | 49.6 | 47.3 |
>
> In terms of computational efficiency, even though the number of novel samples is small, the base set is large, thus retraining is expensive. In a scenario where an ImageNet-scale dataset is the base set, training a ResNet-50 from scratch usually takes weeks on a single GPU.
>
> ---------------
> [1] Zhu, Kai, et al. "Self-Promoted Prototype Refinement for Few-Shot Class-Incremental Learning." CVPR 2021.
>
> [2] Yoon, Sung Whan, et al. "Xtarnet: Learning to extract task-adaptive representation for incremental few-shot learning." ICML 2020.

---

> ### Author Response · Authors · 2021-11-29
> **Checking in**
>
> Dear reviewer,
>
> Since today is the last day for discussions, if you have other concerns that are not answered below, kindly let us know and we will do our best to address them.
>
> Thank you.

---

### Official Review · Reviewer_qioC · 2021-11-03

**Correctness:** 4
**Technical Novelty And Significance:** 4
**Empirical Novelty And Significance:** 4
**Recommendation:** 8
**Confidence:** 4

**Main Review:**

**Strong points**
Very thorough experiments. I liked the visualizations in Figures 2 and 3. This gives a very good understanding of the extent of the model and how it performs. I also liked the idea of adding classes in each session. The method is novel and intuitive for solving this problem. Finally, the paper is easy to follow and well-structured.

**Weak points**
This method does not work if the feature extractor cannot do a good job. An important part of training a new classifier is to learn the feature extractors. This approach misses this in both ways. First of all, when new classes are introduced, their features are not added to the feature extractor. Second, if the feature extractor does not focus on an important feature in a future class, the method is expected to fail.

**Review**
Based on strong points and weak points, I vote for acceptance of the paper. I think it would be great if the authors can explore the points I mentioned in the weak points, but still, the paper itself is an interesting idea with very good experiments. In addition the idea of using text for further improving the regularization kind of weighted my decision more toward acceptance. It shows that there are interesting ways to develop this method.

**Summary Of The Paper:**

This paper leverages a simple but effective regularization technique for few-shot incremental class learning. The assumption is that at the first stage we have a lot of data and can train a good feature extractor and a network that can classify all the classes to a good degree. Given a few examples of a new unseen class, the authors enforce that the weights of the final layer of the classifier are close to the space that is spanned by the current classifier's last layer weights. They achieve state-of-the-art results on this very challenging task.



**Summary Of The Review:**

The paper proposes a method for using regularization on the final layer of a classifier to improve the performance on novel unseen classes with very few examples. Then delves into the proposed idea and studies it with very good and justifiable experiments and visualizations. As a result, I vote for acceptance.

---

> ### Author Response · Authors · 2021-11-16
> **Response to Reviewer qioC**
>
> Thank you for your positive recommendation!
>
> **Re1:** “Keeping the feature extractor fixed.”
>
> Past work in FSCIL emphasized the importance of eliminating a need for retraining/fine-tuning the feature extractor (Chen & Lee 2021, Ren et al. 2019). However, as you have noted, there may be scenarios where the initial feature extractor is inadequate; e.g. in cases of domain mismatch between base and novel classes. In very early experiments, we tested a version of our approach with a fine-tuned feature extractor, but found that performance on base classes was poor due to catastrophic forgetting of features useful for base classes. Penalizing the change in the feature extractor while setting them trainable did not effectively remedy the situation. However, an interesting future direction is to explore effectiveness of training residual adapter modules [1] in between layers while keeping the rest of the model fixed which would facilitate feature learning from novel classes.
>
> --------------------
> [1] Rebuffi, Sylvestre-Alvise, Hakan Bilen, and Andrea Vedaldi. "Learning multiple visual domains with residual adapters." In Proc. of NIPS 2017.

---

### Official Review · Reviewer_EoLF · 2021-11-03

**Correctness:** 3
**Technical Novelty And Significance:** 3
**Empirical Novelty And Significance:** 2
**Recommendation:** 5
**Confidence:** 4

**Main Review:**

**Strengths**

- This paper attacks a very important and practical challenge. and the motivation of the proposed method is well presented.

- The idea of regularizing the novel weight vector using the base ones to prevent the network from capturing spurious correlations seems interesting.

**Questions**

- I'm a bit confused by the regularization term imposed on the old class $R_{old}$. In the following session, since the feature extractor is fixed and I assume there are no available samples from the base classes, why the weight vectors of base classes are not fixed? And why this regularization term is important?

- While better quantitative results are reported in the experiment section, it remains unclear to me why restricting the novel feature vectors to be in the subspace of base classes can improve the overall performance. This motivation opposes some prior work on subspace-motivated continual learning framework, e.g., [1]. My main concern is why this regularization will not cause interference to base and other incremental classes, as the classification boundaries are all restricted to the subspace defined by the base classes, and I do not see any terms used to encourage a robust subspace obtained by the base classes only.

- Some visualizations, e.g. to the weight vectors are expected to further support the effectiveness of the proposed regularization terms, especially the semantic subspace regularization.

- Missing discussions on some recent work, e.g., [2,3].

**Minor**

This paper demands additional rounds of proofreading.

Non-exclusive examples:

Page 5 line 7 double 'append'

Section 2 paragraph 3 'learners.Existing' missing space

Section 3 paragraph 4 'Qi et al. (2018) initialize novel' -> 'initializes'

Section D.1 line 2 'with learning starting' -> 'with a learning rate starting'

[1] Continual Learning in Low-rank Orthogonal Subspaces, NeurIPS, 2020
[2] Few-Shot Incremental Learning with Continually Evolved Classifiers, CVPR, 2021
[3] Self-Promoted Prototype Refinement for Few-Shot Class-Incremental Learning, CVPR, 2021

**Summary Of The Paper:**

This paper introduces a new method of few-shot incremental learning with its regularization-based method motived by a subspace view.
Regularization terms to the linear layers of incremental classes are proposed to encourage the novel weight vector to be in the subspace spanned by those of the base classes, and semantic information from the text domain can further be leveraged to guide the interpolation within the base-class subspace.

**Summary Of The Review:**

As a simple idea, more discussions and empirical supports are expected to make it more convincing. I'll raise my score if all the concerns are effectively addressed.

---

> ### Author Response · Authors · 2021-11-16
> **Response to Reviewer EoLF**
>
> Thank you for the thoughtful & detailed review! We hope our answers below address your concerns; please let us know if you have any additional questions.
>
> **Re1:** “Why not freeze the base weight vectors?”
>
> We ran experiments in which we froze classifier weights of all previous classes (base + previous novel). This resulted in drops in performance for both the fine-tuning baseline and subspace regularizer in the long run (no memory setting). Intuitively, letting all parameters of the last layer be trainable gives more parameters (expressivity) to fit novel classes. However, more expressivity might cause overfitting which we balance out with subspace regularization.
>
> Results are shown below. In fine-tuning, while fixing older weights results in slight gains in accuracy, it results in worse performance in the long run.
>
> |       Weighted Average                |   0 |    1 |    2 |    3 |    4 |    5 |    6 |    7 |    8 |
> |:------------------------------------|--------:|-----:|-----:|-----:|-----:|-----:|-----:|-----:|-----:|
> | Fine-tune (older weights fixed)     |    80.4 | 62.5 | 50.2 | 39.6 | 30.8 | 22.7 | 16.0   | 12.9 | 10.7 |
> | Fine-tune (older weights not fixed) |    80.4 | 61.8 | 49.9 | 40.5 | 34.0   | 31.6 | 28.4 | 27.9 | 26.5 |
> | Subspace Reg. (older weights fixed)     |    80.4 | 70.6 | 62.7 | 56.2 | 50.8 | 46.0   | 42.1 | 38.4 | 35.3 |
> | Subspace Reg. (older weights not fixed) |    80.4 | 71.7 | 66.9 | 62.5 | 58.9 | 55.0   | 51.9 | 49.8 | 46.8 |
>
>
> With respect to the significance of $R_{old}$; when all classifier weights are trainable, we provide experiments in Table 3 in the appendix where we discuss implications for removing $R_{old}$. In summary, $R_{old}$ is useful throughout all the sessions of semantic subspace regularizer while it becomes more important in the later sessions with simple fine-tuning.
>
> **Re2:** “Why regularization of novel class weights towards the base subspace will not cause interference to base classes?”
>
> We observed that most of the prediction errors in baseline models are due to recency bias [1]: the most recently learned classes are assigned disproportionately large probability relative to over old classes. Subspace regularization helps address this issue.
> We did not observe interference with base classes to be a major issue. This may be partly due to the fact that regularization enforces only a soft subspace constraint rather than a hard one (final novel weights do not necessarily lie on the exact subspace; rather close to it).
>
> **Re3:** “Visualizations of weight vectors.”
>
> In the appendix, Fig. 5 shows a trajectory of the classifier weights based on the first two principal components. As an example, the semantic target for the novel class crate falls close to the base class barrel, so the classifier weight $w_{crate}$ is attracted towards its corresponding target.
>
> **Re4:** “Missing discussions to previous benchmarks in CVPR 2021.”
>
> Thank you for pointing out these references, they are added to the revision now.
>
>
> ----------------------------
> [1] Masana, Marc, et al. "Class-incremental learning: survey and performance evaluation on image classification." arXiv preprint arXiv:2010.15277 (2020).

---

> ### Author Response · Authors · 2021-11-29
> **Checking in**
>
> Dear reviewer,
>
> Since today is the last day for discussions, if you have other concerns that are not answered below, please let us know and we will do our best to address them.
>
> Thank you.

---

### Decision · Program_Chairs · 2022-01-20

**Decision:**

Accept (Poster)

**Comment:**

The paper proposes a subspace regularization technique that encourages the new class weight vector to be in the subspace spanned by those of the base classes for few-shot class incremental learning. Even though similar techniques exist in few-shot learning literature, reviewers appreciate the simplicity of the method and thorough experiments. The authors have revised the paper to include missing references suggested by reviewers during the rebuttal. They were not able to add experiment comparisons to Tao et al. (2020) and Chen & Lee (2021) as requested by reviewer Vrap due to missing code release.  Please consider adding them in your draft later.